# Reinforcement determines the timing dependence of corticostriatal synaptic plasticity in vivo

Simon D. Fisher[1], Paul B. Robertson[2], Melony J. Black[1], Peter Redgrave[3], Mark A. Sagar[2], Wickliffe C. Abraham[4] & John N.J. Reynolds[1]

Plasticity at synapses between the cortex and striatum is considered critical for learning novel actions. However, investigations of spike-timing-dependent plasticity (STDP) at these synapses have been performed largely in brain slice preparations, without consideration of physiological reinforcement signals. This has led to conflicting findings, and hampered the ability to relate neural plasticity to behavior. Using intracellular striatal recordings in intact rats, we show here that pairing presynaptic and postsynaptic activity induces robust Hebbian bidirectional plasticity, dependent on dopamine and adenosine signaling. Such plasticity, however, requires the arrival of a reward-conditioned sensory reinforcement signal within 2 s of the STDP pairing, thus revealing a timing-dependent eligibility trace on which reinforcement operates. These observations are validated with both computational modeling and behavioral testing. Our results indicate that Hebbian corticostriatal plasticity can be induced by classical reinforcement learning mechanisms, and might be central to the acquisition of novel actions.

[1] Department of Anatomy and the Brain Health Research Centre, Brain Research New Zealand, University of Otago, Dunedin 9054, New Zealand. [2] Laboratory for Animate Technologies, Auckland Bioengineering Institute, University of Auckland, Auckland 1142, New Zealand. [3] Department of Psychology, University of Sheffield, Sheffield S1 1HD, UK. [4] Department of Psychology and the Brain Health Research Centre, Brain Research New Zealand, University of Otago, Dunedin 9054, New Zealand. Correspondence and requests for materials should be addressed to J.N.J.R. (email: john.reynolds@otago.ac.nz)

Synaptic plasticity, the change in strength of connections between neurons, is a key substrate for learning. A prevailing model of naturally occurring plasticity, termed spike-timing-dependent plasticity (STDP), describes how synaptic weights can be changed in a physiologically relevant manner by precisely controlling the timing relationship between pre and postsynaptic activity[1]. In most brain areas STDP functions in the expected 'Hebbian' sense[2], where presynaptic activity that is followed within milliseconds by postsynaptic firing (positive timing) potentiates synaptic efficacy, presumably by signifying that the input contributed to the cell firing. Conversely, when presynaptic activation occurs after postsynaptic activity (negative timing) the synapse is depressed, signifying that arrival of the input was not causally related to postsynaptic cell firing.

Plasticity between the cortex and the striatum has long thought to be critical for motor control and learning[3, 4]. Investigations of STDP at corticostriatal synapses have reported bidirectional plasticity following both positive and negative pairings with wide variation in results[5–7]. Almost exclusively, these studies have been performed in the controlled environment of brain slice preparations in vitro. A disadvantage of this paradigm is that the signals responsible for evoking the neural plasticity are uncoupled from behaviorally relevant signals that are likely to occur in vivo. During the learning of action-outcome associations, a critical component of basal ganglia function[8], multiple signals are presented at different times to the input nucleus, the striatum. These include inputs representing action initiation and execution, and

those associated with the sensory consequences of performed actions. The latter, including phasic dopamine responses to novel and rewarding events[9], are thought to operate as critical reinforcement signals. Thus, it has been proposed that a conjunction of activity at specific corticostriatal synapses, and the firing of striatal spiny projection neurons (SPNs) that accompanies action selection and execution, is thought to set-up an eligibility trace that decays over time[10]. It is within this eligibility time window that a phasic dopamine reinforcement signal must arrive to credit the correct synapses with causing the outcome[8]. Although found to be critical for bidirectional corticostriatal STDP[6, 7], dopamine is typically released in brain slice preparations as a by-product of corticostriatal activation, rather than arriving sometime later, as would occur naturally with behavioral reinforcement. Moreover, the delayed phasic release of dopamine onto striatal SPNs and interneurons in vivo is accompanied by glutamatergic input from the thalamus, both of which are evoked by the same sensory events[11–13]. Thus, when considering the absence of temporally relevant reinforcement timing relationships in brain slice studies, together with variation in tonic GABA receptor activation[5, 6, 14], the wide variation in STDP results reported is understandable. Consequently, there is a critical need for an experimental paradigm to study corticostriatal plasticity that incorporates a full range of physiologically relevant signals that can be delivered with appropriate timings.

At present, it is unknown whether corticostriatal STDP pairings alone can induce bidirectional plasticity in intact animal

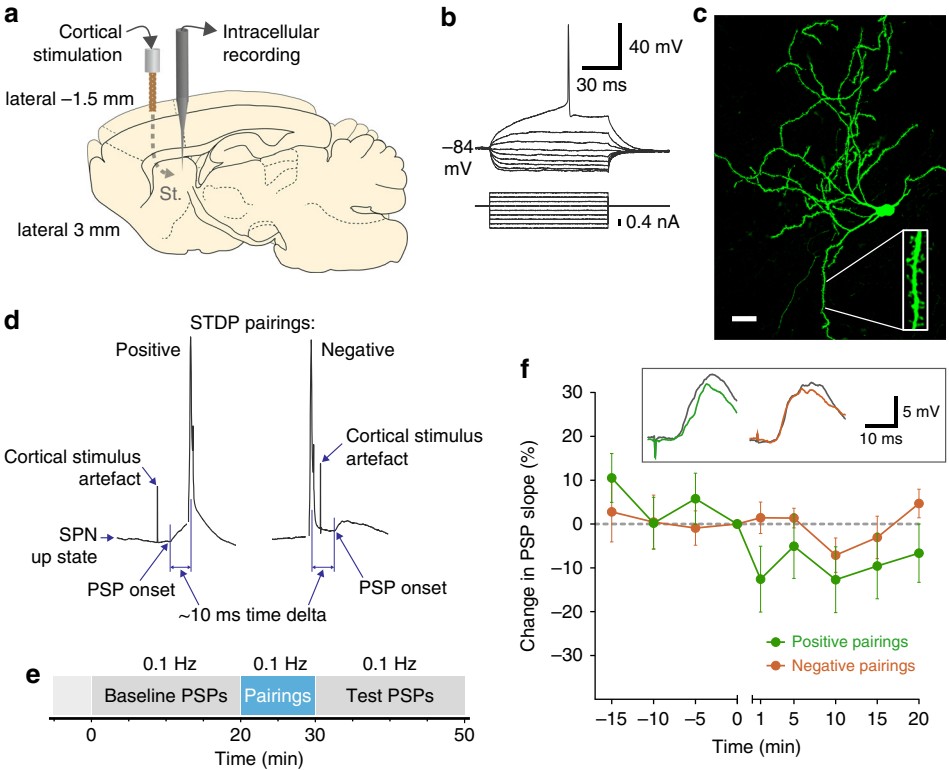

**Fig. 1** Corticostriatal STDP in vivo preparation. **a** Stimulation and recording locations in rat brain. Cortical stimulation is contralateral to striatal recording. St striatum. **b** Current–voltage relationship illustrating characteristic SPN responses, including inward rectification and ramp potential. **c** Biocytin-filled recovered SPN in the dorsal striatum, with inset showing spines (enlarged 300%). *Scale bar* = 20 μm. **d** STDP timing arrangement for positive and negative pairings, delivered at 0.1 Hz in each of the experimental protocol phases depicted in **e**. PSP, postsynaptic potential. **e** Experimental protocol phases.
**f** Percentage change in PSP slope measures induced by positive (*green*, n = 6) and negative (*orange*, n = 7) STDP pairings (Linear mixed-effects model [LME] for all points post pairing between protocols, see Methods section, estimated effect size = 8.8 ± 7.1%, $F_{1,10}$ = 1.54, P = 0.24). Baseline measures are plotted on the initial portion of the X axis, no measures are recorded during plasticity protocol, and the test PSP measures are plotted on the second portion of the X axis. *Inset* shows representative examples of one minute averaged PSPs from the end of the baseline (*gray*) and the end of the test period (color of experimental group). Data presented as mean ± s.e.m

preparations. Nor is it clear how physiological reward-related signals in vivo, including phasic dopamine, and their timing may influence the induction of STDP. Answers to these questions would have significant implications for understanding how basal ganglia plasticity contributes to normal reinforcement learning

processes responsible for action discovery and the biasing of action selection[8, 15]. Such an understanding may also be necessary before the dysfunctional learning that occurs in conditions such as obsessive-compulsive disorder[16] and substance dependence[17] can be fully appreciated.

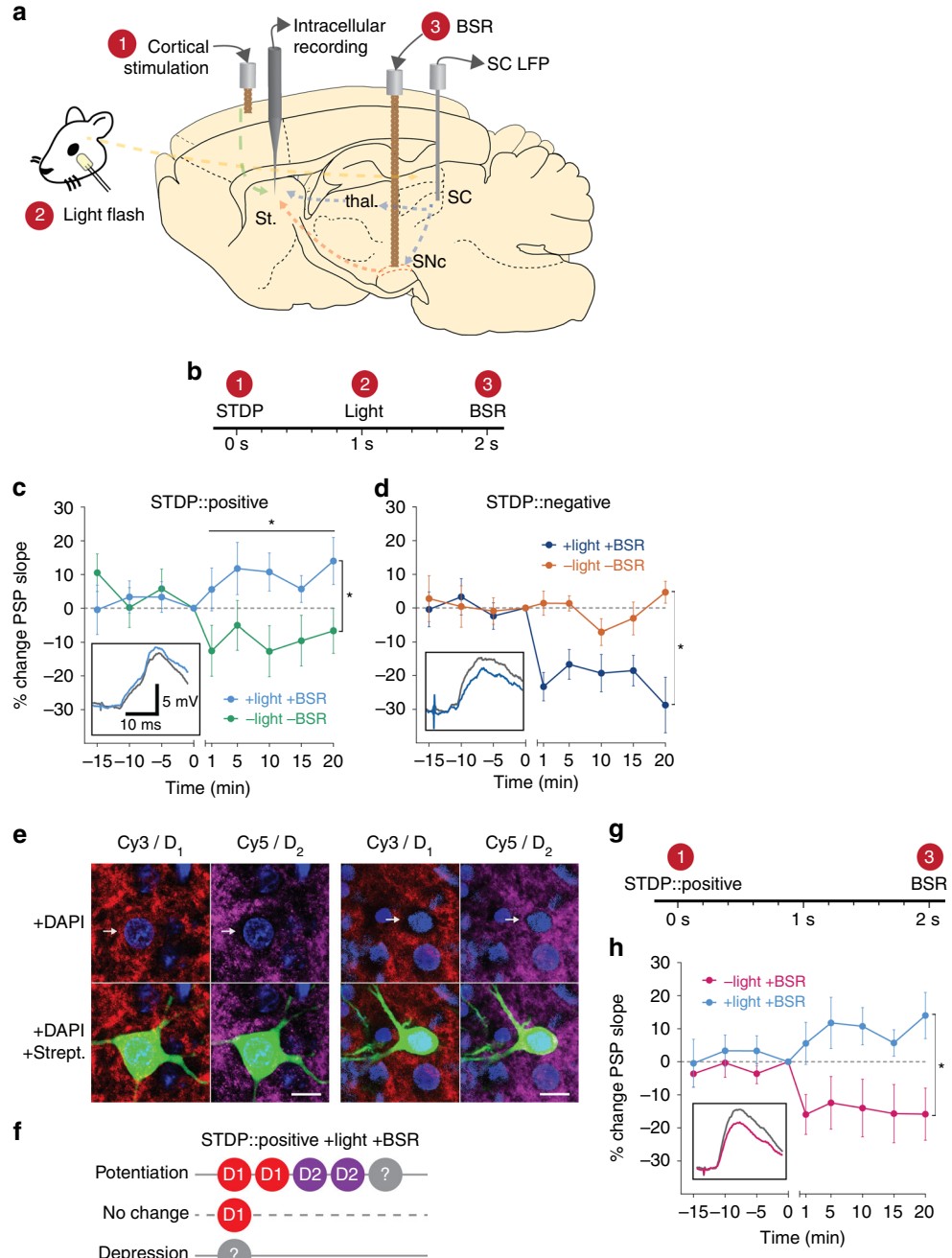

**Fig. 2** Reinforcement modulates corticostriatal STDP to enable bidirectional plasticity. **a** Schematic view of stimulating and recording sites. Numbers in *red* relate to components of the experimental protocol **b**. **c** Positive STDP pairings are modulated by the reward components (*blue*, n = 7) to induce lasting potentiation of corticostriatal synapses (time points after protocol vs. baseline, LME, est. effect size = 8.8 ± 3.3%, $F_{1,46}$ = 6.69, P = 0.013), which was significantly different from positive pairings without reinforcement (LME, est. effect size = 20.1 ± 8.8%, $F_{1,11}$ = 5.20, P = 0.043). *Inset* shows representative PSPs as defined in Fig. 1(f). **d** Negative STDP pairings are modulated by the reward components (*dark blue*, n = 5) to induce lasting depression of corticostriatal synapses, significantly different from negative pairings without reinforcement (LME, est. effect size = 24.5 ± 8.3%, $F_{1,10}$ = 10.40, P = 0.009). **e** Examples of $D_1$ (Cy3) and $D_2$ (Cy5) positive SPNs intracellularly filled with biocytin and visualized with DyLight 488-conjugated streptavidin. Note the presence of double labeling indicated by a perisomal ring within the cell membrane. *Scale bar* = 10 μm. **f** Summary dopamine receptor results for the positive STDP pairings with reinforcement group, indicating a mixed receptor expression in cells that potentiated. **g** Modified plasticity protocol without the light flash. **h** Positive STDP pairings, when modulated by BSR alone at 2 s (*pink*, n = 5), induce long-lasting depression, significantly different from positive pairings with light and BSR (LME, est. effect size = 24.4 ± 9.6%, $F_{1,10}$ = 6.40, P = 0.029). Data presented as mean ± s.e.m

Here, we report an experimental paradigm in which bidirectional, Hebbian corticostriatal STDP can be achieved in vivo, but only through modulation by physiologically and behaviorally relevant reinforcement signals involving dopamine and adenosine signaling. We show that a secondary reinforcing light stimulus, conditionally reinforced by phasic dopamine cell activation, induces potentiation when delivered within a behaviorally relevant time window following positively timed STDP pairings, and robust depression when delivered following negatively timed pairings. The timing of the secondary reinforcing signals is critical, as plasticity outcomes are abolished if they are presented outside a critical, behaviorally relevant eligibility window. These findings suggest an important reinforcement role for a conditioned sensory stimulus in modulating corticostriatal motor plasticity in a time-dependent manner. Manipulations of the relevant reinforcement components in a computational model and a novel behavioral task proceeded to demonstrate how such plasticity processes could relate to the discovery of novel actions.

## Results

**Corticostriatal STDP in vivo.** Spike-timing-dependent plasticity is typically induced when presynaptic activation occurs within a time window of 5–30 ms around the time of postsynaptic cell firing. To evaluate corticostriatal STDP in vivo, positive (pre–post) and negative (post–pre) STDP pairings were constructed by pairing electrical stimulation of the motor cortex with intracellular current injections, suprathreshold for action potential generation in striatal SPNs, in urethane-anesthetized rats (Fig. 1a–d). STDP pairings were performed during the beginning of the SPN membrane potential 'up state' close to firing threshold, due to the likelihood of enhanced $Ca^{2+}$ signals in this relatively depolarized phase[18]. Importantly, placement of the stimulating electrode in the intact cortex remote from the striatum ensured that we did not have concomitant dopamine release directly due to the stimulation. We first tested the effect of STDP pairings (0.1 Hz for 60 repetitions, Fig. 1e) delivered in the absence of reinforcement on corticostriatal postsynaptic potentials (PSPs). Positive pairings induced minor depression of the synaptic response, while negative pairings caused no significant change (Fig. 1f); these groups did not differ significantly from each other. While these results agree in part with some in vitro data, in that positive pairings tended toward depression[5, 14, 19], overall, these conventional corticostriatal STDP protocols in vivo were not able to induce the robust, bidirectional changes in synaptic efficacy observed in vitro.

**Corticostriatal STDP is modulated by sensory reinforcement.** It has been proposed that plasticity at corticostriatal synapses is fundamental to learning and cognitive processes involved in action discovery and selection[8]. Behaviorally relevant reinforcement signals might therefore be expected to influence the induction of plasticity at these synapses through dopamine modulation[20, 21]. Therefore, we tested the effect of a reinforcing visual stimulus on STDP, by applying a light flash to the rat's eye during the pairing protocol. Previously, we have found that under anesthesia, a light stimulus fails to evoke either a phasic dopamine response[11] or an up state membrane transition in SPNs[22] in the striatum. Both of these effects occurred only when the light flash was applied following local injection of the $GABA_A$ receptor antagonist bicuculline into the midbrain superior colliculus (SC), to overcome the suppressive effects of anesthesia. In the present study, we did not use a pharmacological means to restore collicular sensory processing. Instead, we used a process of classical conditioning, in which a light flash to the rat's contralateral eye was paired with an unconditioned brain stimulation reward

(BSR). We reasoned that this would ultimately result in the light flash becoming a conditioned reinforcer and itself driving dopamine release into the striatum.

To test the effects of sensory reinforcement within timescales relevant to behavioral conditioning, we followed STDP pairings with a light flash after a delay of 1 s (Fig. 2a, b), while establishing the secondary reinforcing properties of the light flash by applying BSR a further 1 s after the light flash. The BSR was electrical stimulation of the substantia nigra pars compacta (SNc) using parameters known to be behaviorally reinforcing[21] and to release dopamine into the striatum[23]. Together, the excitatory STDP event followed by the light flash and BSR model a behavioral scenario in which a motor action causes the appearance of a sensory event made salient by association with an unconditioned reward. This should lead to reinforcement of this action-outcome association in the striatum.

When the conditioned light stimulus was applied after positive STDP pairings, significant lasting potentiation of corticostriatal synapses was induced, in sharp contrast with the depression seen without reinforcement (Fig. 2c). Conversely, the same reinforcement components applied to negative STDP pairings induced robust depression (Fig. 2d). Of the neurons in the positive STDP reinforced group that could be recovered post-mortem and their dopamine receptor expression reliably determined ($n = 5$, see Methods section), both $D_1$ and $D_2$-expressing SPNs exhibited potentiation (Fig. 2e, f). Additionally, a single neuron recovered from the negative STDP reinforced group that exhibited depression expressed $D_1$ dopamine receptors. These findings suggest that potentiation in the positive group and depression in the negative group was not determined solely by the specific SPN's dopamine receptor complement. Since the reinforcement administered was the same in both groups, the differential outcomes observed depended solely on the order of the initial STDP pairing. Thus, corticostriatal STDP in vivo, induced with millisecond timing, was strongly modulated by a discrete reinforcement event that occurred on a much longer, behaviorally relevant timescale of one second.

**The role of the conditioned light stimulus in STDP.** We designed the STDP reinforcement protocol so the light flash would act as a conditioned reinforcing stimulus, through its reliable association with BSR. It is possible however that the dopamine released by the BSR itself may have been an essential contributory factor. To determine whether the secondary reinforcing properties of the light were critically important, we tested a protocol in which the BSR was delivered, without the preceding light flash, 2 s following the STDP pairing (Fig. 2g). BSR alone failed to induce potentiation with positive STDP pairings (Fig. 2h). Indeed, the outcome was indistinguishable from when positive STDP pairings were administered without any form of reinforcement (c.f. Fig. 1f). This result verified the essential secondary reinforcing role played by the light flash in inducing potentiation after positive STDP timing. It also provides an important insight concerning the duration of the period during which the STDP is eligible for 'reinforcement'—reward-related signals with a 2 s delay were not capable of modulating striatal plasticity.

To investigate further the role of light reinforcement in corticostriatal STDP, we measured the membrane response of recorded SPNs to each visual stimulus as it was delivered during the plasticity protocols. Previously, we have shown that visual stimuli can depolarize the membrane of SPNs to up state levels if the light was delivered in conjunction with local pharmacological disinhibition of the deep layers of the SC[22]. In this state, retinal projections to the SC superficial layer were able to activate deep

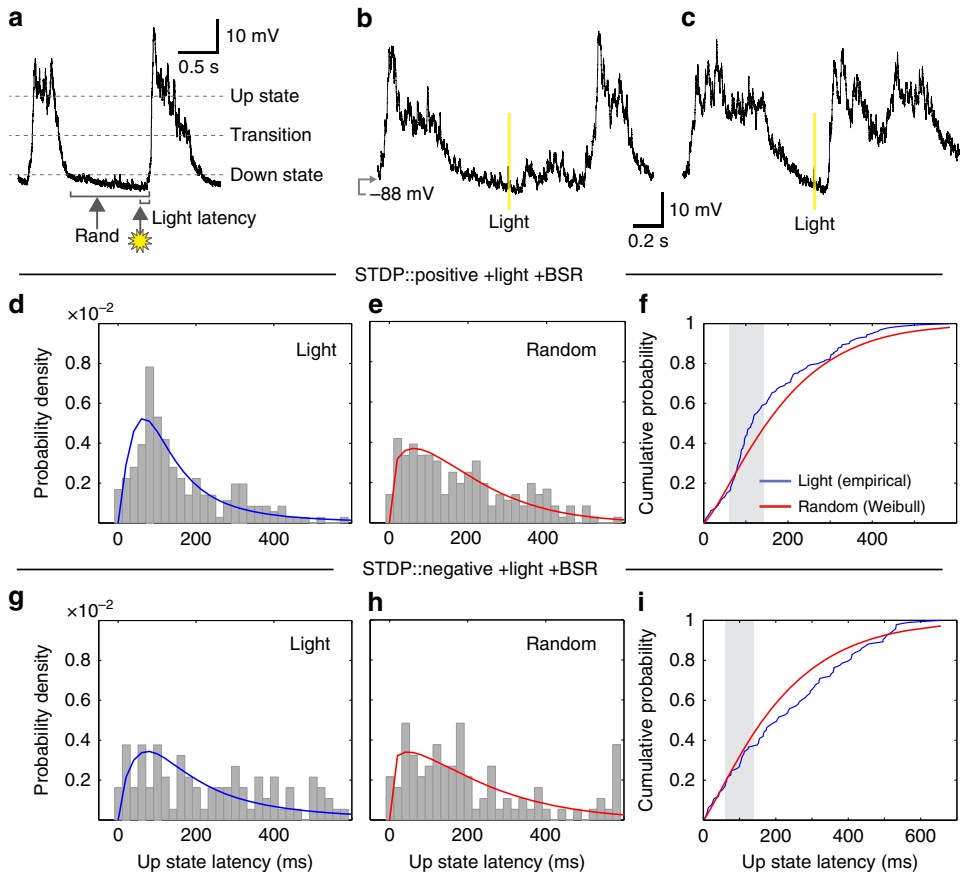

**Fig. 3** Modulation of STDP by the conditioned light stimulus is associated with increased probability of up state initiation in SPNs. **a** Latencies to the up state (detected using the transition level indicated) were measured for both a random event in the down state and from the light event. **b** Representative example of instances in which a light event failed to initiate an up state and **c** when an up state was initiated. **d**, **g** Probability distribution function (PDF) of latencies to the next up state following all light events plotted for positive ($n = 177$; **d**) and negative ($n = 94$; **g**) pairings groups, and fitted with a log-logistic distribution curve. **e**, **h** For comparison, the PDF of latency to the up state from a random point in the same down state was calculated for both groups and fitted with a Weibull distribution curve. The light-evoked latencies for the positive pairings group were significantly different from, and hence unlikely to have been sampled from, the random latency distribution (Kuiper's test, $P < 0.00001$; although they were not significantly different in the negative pairings group ($P = 0.16$). **f**, **i** Cumulative distribution functions (CDF) for the light and random events were constructed, and the probability of an up state occurring at the expected interval of 60–140 ms (shaded region) following the light was calculated from the CDFs by subtracting the probability of an up state occurring during this period randomly from the probability of an up state occurring during this period following a light event. In the positive pairings group (+0.13), but not the negative pairings group (+0.03), light events showed a notable probability above chance of initiating an up state

layer collicular neurons, thereby providing phasic excitatory drive to striatal micro-circuits via the intralaminar thalamic nuclei[22]. Here, we hypothesized that repeated associations between the light flash and BSR would potentiate the tecto-thalamostriatal projection sufficiently to enable the flash to drive striatal SPNs to the up state, without the necessity of collicular disinhibition[22].

To test this, we compared the latency to an SPN up state following a light flash with the latencies of spontaneously-occurring up states (Fig. 3a–c). We found that the probability of an up state transition following a conditioned light flash was significantly greater than would be expected by chance. Importantly, the increase in SPN up state probability was observed for the protocol that subsequently induced corticostriatal potentiation, i.e., positive STDP pairing followed by reinforcement (Fig. 3d–f). The observed changes in the up state latency distribution were similar to those found previously when the SC was pharmacologically disinhibited[22]. Conversely, the negative timing reinforced group showed a trend toward decreased SPN excitation by the light (Fig. 3g–i). These group differences developed within the first 3 min (18 trials) of pairing, despite both groups receiving the same reinforcement protocol. Thus, in the absence of any relevant group differences in SPN

cellular properties (Supplementary Fig. 1), differences between groups in light-induced up state probability must have been determined by the order of STDP pairing, and may therefore result from a plastic mechanism localized to the striatum. One possibility is that the increase in light-induced up state transition in the positive pairings plus reinforcement group may reflect potentiation of thalamic synapses onto SPNs, with depression of these inputs induced by negative pairings.

Taken together these data suggest that the light flash that followed the STDP, and preceded the BSR by 1 s, was necessary to modulate positive STDP pairings and induce corticostriatal potentiation. Since the association between the light and BSR seemed to have an enhancing effect akin to that seen following local collicular disinhibition, it is likely that the light had been conditioned by the BSR to facilitate tecto-thalamic drive to striatal spiny neurons. Disinhibited deep layer collicular neurons also simultaneously drive midbrain dopamine neurons via their axon collaterals, to deliver phasic dopamine into the striatum[8, 11]. Thus, the light may also have provided a phasic dopamine signal to the striatum, in conjunction with thalamostriatal depolarization, within the critical time period following the STDP pairing. Our next experiments tested the role of neuromodulators likely to

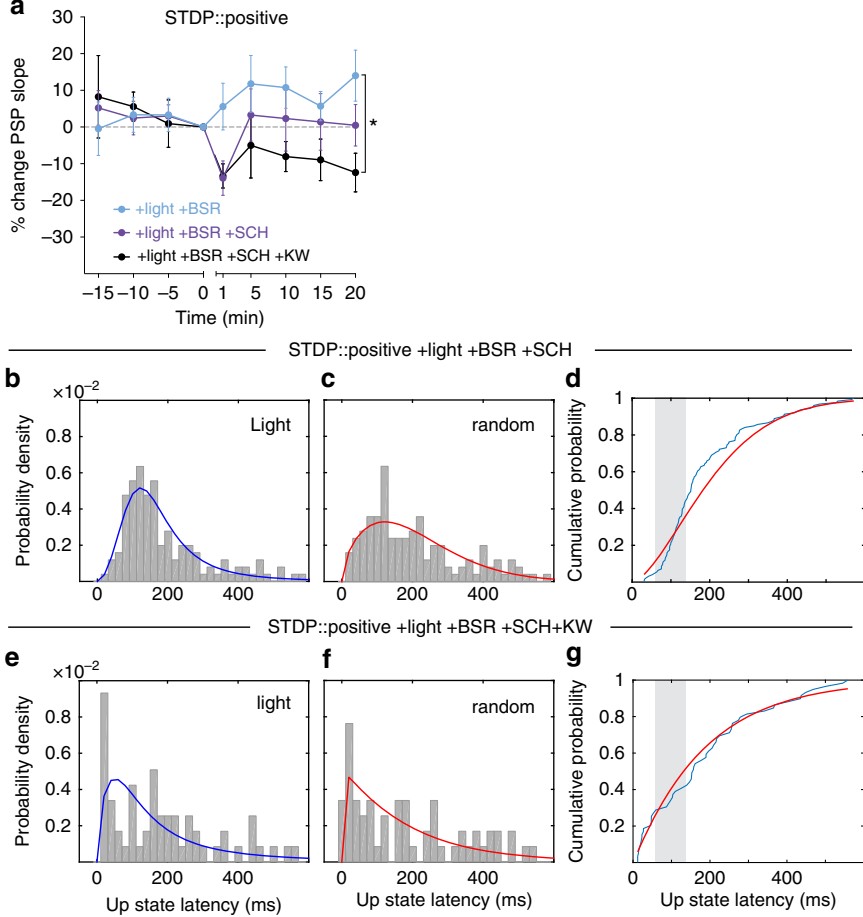

**Fig. 4** The effect of the conditioned light stimulus on potentiation is mediated by dopamine $D_1$ and adenosine $A_{2A}$ receptor signaling. **a** Systemic administration of SCH 23390 (+SCH, $n = 6$) partially abolished the potentiation previously found with the positive timing STDP protocol with reinforcement. Combined SCH 23390 and KW-6002 (+SCH +KW, $n = 4$) systemic administration resulted in robust depression of corticostriatal synapses in comparison to the reinforced positive timing group (LME, est. effect size = $18.5 \pm 7.1\%$, $F_{1,9} = 6.69$, *$P = 0.02$). Data presented as mean $\pm$ s.e.m. The PDFs for the +SCH group (**b**, **c**) and +SCH +KW group **e**, **f** were calculated with the same methodology outlined in Fig. 3. The light-evoked latencies for the +SCH group ($n = 124$) were significantly different from, and hence unlikely to have been sampled from, the random latency distribution (Kuiper's test, $P < 0.00001$); light-evoked latencies ($n = 58$) were not significantly different in the +SCH +KW group (Kuiper's test, $P = 0.09$). **d**, **g** In the +SCH group (+0.18), but not in the +SCH +KW group (+0.02), the light events showed a notable probability above chance of initiating an up state

have been released by the light in the induction of corticostriatal potentiation following positive STDP pairings.

**Conditioned light modulates STDP via dopamine and adenosine**. Dopamine signaling via $D_1$ receptors has been associated with second messenger systems in SPNs favoring synaptic potentiation[7], [24]. To test whether $D_1$ dopamine receptor signaling is required for the potentiation found when positive STDP pairings were modulated by sensory reinforcement, the $D_1$ receptor antagonist SCH 23,390 was administered systemically ~25 min prior to the plasticity protocol. While the group average result of the SCH group showed no net plasticity (Fig. 4a), there was a high degree of variability within the group, with some individual cells still exhibiting potentiation (Supplementary Fig. 2). As $D_1$ receptors are primarily expressed on the direct pathway subpopulation of SPNs[25] and potentiation was also found in $D_2$-expressing SPNs, we tested whether also targeting potentiation pathways in indirect pathway SPNs would result in a more robust blockade of the modulatory effect of sensory reinforcement. Adenosine $A_{2A}$ receptors are primarily expressed on indirect SPNs[26], and due to comparable intracellular signaling cascades, are likely to have a similar role to the $D_1$ receptor in LTP induction in these neurons[7]. Simultaneous

blockade of $D_1$ receptors, and $A_{2A}$ receptors via KW-6002, robustly abolished the potentiation previously found with both receptors intact (Fig. 4a), revealing LTD reminiscent of the effect of positive pairings alone (c.f. Fig. 1f). Importantly, there was no difference in the PSP prior to the plasticity protocol between the no-drug and the +SCH/+KW group, indicating a minimal blockade of presynaptic $A_{2A}$ receptors ($P = 0.1$, unpaired, two-tailed $t$ test) which could have confounded the results by reducing presynaptic glutamate release[27]. These results suggest that the potentiation induced by positive STDP pairings modulated by sensory reinforcement relies on both $D_1$ and $A_{2A}$-mediated mechanisms, likely expressed on a mixed population of postsynaptic SPNs.

Consistent with our earlier supposition that the protocols may have also induced plasticity affecting thalamostriatal inputs to SPNs, $D_1$ receptor blockade, which failed to block potentiation in all neurons, also did not block the increase in up state probability in response to the light flash (Fig. 4b–d). This suggests that dopamine $D_1$ signaling alone is not essential for this form of thalamostriatal plasticity. However, full block of potentiation by $D_1$ and $A_{2A}$ antagonists also fully blocked the increase in light-induced up state probability (Fig. 4e–g), suggesting a complex interrelation between neuromodulator-induced corticostriatal and thalamostriatal plasticity requiring future research.

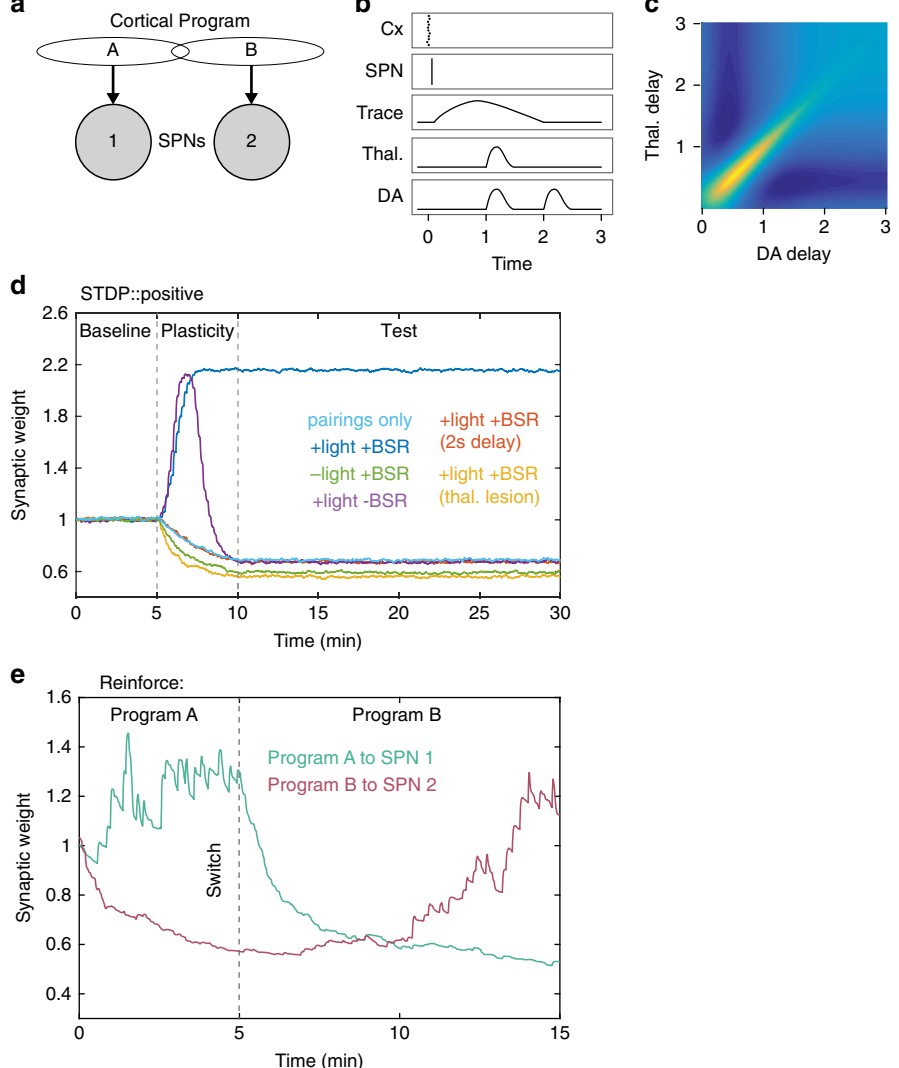

**Fig. 5** Computational model illustrates reinforced STDP mechanisms and how they can support selection of actions. **a**, **b** An overview of the model components, and the formation of an eligibility trace, based on the in vivo protocol (see Methods section for details). **c** Illustration of the synaptic weight change as a factor of dopamine and thalamic signal timing, indicating a time delay optimal for potentiation for both components of between ~0.4 and 1.1 s. **d** Model recreation of in vivo plasticity protocol conditions using positive pairings: pairings alone (*light blue*), with reinforcement (*dark blue*), and BSR delivery only without light (*green*). Also modeled predictions of untested conditions: light only without BSR (*purple*), which as expected induces initial potentiation that becomes depression due to habituation of the light when continually presented without primary reward association; both reinforcement components delayed by 2 s instead of one (*red*) and also with thalamic input 'lesioned' (*orange*). Baseline period involves no STDP pairings or reward delivery, reinforcement is delivered during the plasticity period, and synaptic weights are tested again during the test period. **e** Motor program A was reinforced during the first 5 min, at which time a switch to reinforcement of motor program B occurred

**Computational model of corticostriatal STDP.** To validate and explore these STDP reinforcement findings further we created a computational model of corticostriatal interactions, based on a previous spiking neuron model that incorporated the timing of action and reinforcement signals[10]. The model considers the interaction of reinforcement signals with an eligibility trace[10], and includes updated rules of reinforced STDP in vivo that apply to both direct and indirect SPN types (Fig. 5a–c).

With this model, we estimated changes in synaptic efficacy throughout the pairing period that were not able to be accurately measured from our experiment because of the influence of the depolarized up state and superimposed spike on the PSP slope. It also allowed us to test experimental conditions that we hypothesized would not induce potentiation, so as not to unnecessarily extend the ethical cost of these difficult and low

yield experiments. First, we found all modeled results to be consistent with key in vivo findings (Fig. 5d), namely the requirement for appropriately timed reinforcement components to induce synaptic potentiation following positive pairings. Next, our model reported that when the light is presented alone without BSR, an initial rise in potentiation is seen during the pairing protocol due to the novelty effect of the light in releasing dopamine[28]. However, the effect of the light rapidly habituated without paired primary reward, as response habituation is the default mode in the deep collicular layers[29]. Thus, corticostriatal depression was induced with ongoing positive pairings in the presence of a diminished light response (Fig. 5d). Additionally, when the light and BSR combination was delayed 2 s from the STDP pairings, or the thalamostriatal pathway was 'lesioned', a depression outcome similar to positive pairings alone was induced.

To explore potential functional roles of reinforced corticostriatal STDP, the model utilized two cortical motor programs (Fig. 5a; see Methods section). Synchronous activation of motor program A was induced, and then reinforcement was delivered contingent with program A activity (Fig. 5e). After 5 min a switch occurred: motor program A activity was no longer reinforced, and instead motor program B was reinforced. The differential change in mean synaptic weights of the two programs represents the extinction of a previous motor program and discovery of a new program, suggesting that this form of plasticity could underlie action discovery processes.

**Action discovery is consistent with corticostriatal STDP**. Potentiation of motor cortex inputs to the striatum has been proposed as a major substrate underlying the basal ganglia's role in action discovery and selection[8]. We therefore tested whether the reinforcement components found to be critical for corticostriatal potentiation in the electrophysiology and modeling experiments could support action discovery in a behavioral task. A joystick task was used, which has been previously characterized as modeling action discovery[15]. In the switching phase of the task, the unseen 'target area' that the animals must move the joystick into in order to receive reward unexpectedly switches location (Fig. 6a, b). The rats were required to learn through trial and error the action required to move into the new target area. Reinforcement in this task consisted of a light flash whenever the joystick entered the correct area. The flash was followed 1 s later by BSR, thereby matching the reinforcement schedule used in the electrophysiology experiments. Rats were automatically advanced to the next target once they met performance criteria. Dependent measures included the number of blocks (target areas) discovered within a session, and the time spent within each block before reaching the criterion. If the plasticity protocol that induced potentiation in the electrophysiology experiments relates to this action discovery task, it would be expected that manipulations of the protocol that lead to no change or synaptic depression would retard action discovery.

Using the light flash+BSR reinforcement set, rats were successfully discovered the location of the reinforced areas (Supplementary Fig. 3), and were able to discover new locations when the target area was switched (Fig. 6c). When the light flash component of the reinforcement set was omitted following a correct joystick movement into the target location, but BSR still delivered alone 1 s later, learning the action required to find the new target was significantly impaired (Fig. 6d, e). This is consistent with the STDP depression induced during the corresponding electrophysiological experiment (Fig. 2g, h). When the light flash was delivered alone and the BSR omitted, learning was also impaired (Fig. 6d, e), consistent with the modeled results in which a light flash alone produced transient potentiation but then depression (Fig. 5d). Conditions in which light stimuli are repeatedly delivered without associated reward lead to rapid habituation in the deep layers of the SC[11, 29]. Hence, it would be expected that any initial striatal response to the light would rapidly decline and that any associated synaptic plasticity would be similar to positive STDP pairing alone (Fig. 1f). Thus, manipulations of the reinforcement components produced similar outcomes in both behavioral and electrophysiological experiments, demonstrating a potential behavioral role of this reinforcement-induced modulation of corticostriatal STDP.

## Discussion

By examining STDP in the striatum in vivo in the context of behaviorally relevant input signals, we have demonstrated a novel form of STDP that could underpin the role of the basal ganglia in

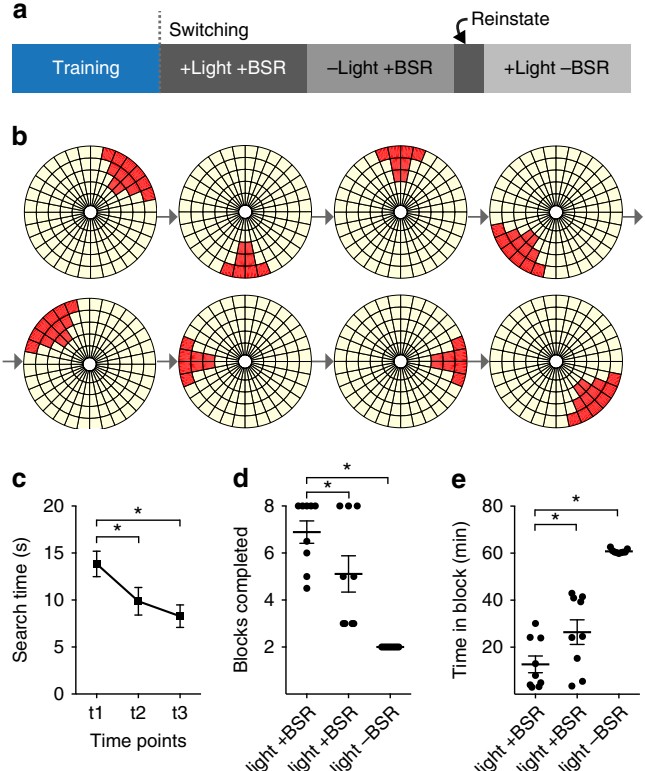

Fig. 6 Learning in a joystick task of action discovery was impaired by manipulation of reward signals. **a** Schematic outline of task phases. Following initial joystick training (Supplementary Fig. 3), switching experiments required rats ($n = 9$) to move the joystick (uncued) in to a series of pseudorandom target areas **b** ('blocks') to elicit a light flash and then BSR one second later. **c** Learning within blocks (one target area) during the switching task is demonstrated by a reduction in the mean search time required to find the target area across normalized time points (RM one-way ANOVA, $F_{1.7,11.6} = 19.2$, $P = 0.0003$; post hoc Dunnett's test, t1 vs. t2, *$P = 0.01$; t1 vs. t3, *$P = 0.0003$). **d** Switching performance with reward manipulations as mean blocks completed (two-way ANOVA, $F_{2,16} = 29.0$, $P < 0.0001$; post hoc Tukey's test, +light +BSR vs. −light +BSR, *$P = 0.03$; +light +BSR vs. +light −BSR, *$P < 0.0001$) and as **e** time spent attempting the target (two-way ANOVA, $F_{2,16} = 68.0$, $P < 0.0001$; post hoc Tukey's test, +light +BSR vs. −light +BSR, *$P = 0.01$; +light +BSR vs. +light −BSR, *$P < 0.0001$). Rats were reinstated with both reinforcement components between manipulations. Data presented as mean ± s.e.m

behavioral reinforcement learning. Neither positively nor negatively timed pairings of pre and postsynaptic activity alone were sufficient to induce reliable corticostriatal plasticity. Instead, we report for the first time in the intact animal that appropriately timed reinforcement signals were required to induce corticostriatal potentiation with positive STDP pairings and depression with negative pairings. Finding that divergent plasticity outcomes were completely dependent on the timing of the pre and postsynaptic activity highlighted the importance of the STDP pairing relationship. The timing requirements for the reinforcement signal were also critical. For positive STDP pairings, potentiation was induced when the initial secondary reinforcement component arrived 1 s following the pairing, but not when the BSR was presented alone 2 s after the STDP pairing. This is the first in vivo evidence to support the existence of an eligibility trace in the context of physiological reinforcement signals.

For reinforcement signals to modulate plasticity with a delay of 1 s after STDP pairing sits in stark contrast to the millisecond

timing constraints typically observed in in vitro plasticity paradigms. Behavioral, physiological and computational evidence consistently implicate dopamine neurotransmission with the timing-dependent reinforcement signal[28, 30, 31]. For example, Yagishita et al.[24] reported that potentiation of SPN spines in vitro was greatest when a phasic dopamine signal occurred approximately 0.6 s after the onset of STDP excitatory activity, but was absent when the arrival of dopamine was delayed by 2 s. This finding supports biophysical models of STDP, which confirm that the dopamine reinforcement signal must arrive within a critical period following a STDP pairing, the putative eligibility trace[10]. These findings are consistent with the present results that suggest there is a decaying eligibility state, within which a reinforcement signal can trigger intracellular consequences that result in altered synaptic efficacy.

The switching of corticostriatal potentiation to depression when the light flash was omitted confirmed the critical role played by the secondary conditioned light for determining the direction of the evoked synaptic plasticity. Once associated with the BSR, the light flash should activate a widely broadcast sustained dopamine signal throughout the striatum[32] by blocking the effects of habituation on the deep layers of the SC[11, 29]. Such an effect would correspond with that seen after light flash induced activation of the tectonigral pathway when the superior colliculus is in a disinhibited state[11]. In the present study, light also induced a depolarization of the recorded SPNs via the glutamatergic tecto-thalamostriatal pathway, likely coincident with the phasic dopamine signal evoked by the same event[8, 22]. Because plasticity such as that shown here is dependent on both dopamine transmission and glutamatergic depolarization[33], it is likely that these two simultaneous phasic, sensory-evoked afferent signals acted in concert to modulate the plasticity induced by corticostriatal STDP in vivo. Brain stimulation reward alone if applied at 1 s might have evoked a suitably timed dopamine signal; however, it may not have provided a coincident depolarization of appropriate magnitude to induce corticostriatal potentiation[21]. Since the rationale for our study was to examine STDP modulation by a conditioned light stimulus, our focus was to test the effect of a light flash, where physiologically co-activated afferent signals converged on the striatum.

The question arises whether our plasticity protocols are relevant to behavioral situations faced by awake animals. First, we used 60 STDP pairings, arguably a large number of trials for simple action/outcome learning, primarily because of its prevalence in the in vitro literature. We thus wanted to bridge between results reported with different preparations. Cui et al.[34] showed that timing-dependent corticostriatal LTP could be biphasic, as determined by the number of pairings (LTP found with 5–15 pairings, no change around 40–50 pairings, and LTP again above 60 pairings). This effect may be specific to their in vitro preparation, or it may reflect unique plasticity processes at corticostriatal synapses. If the latter is true, then potentially we may have found the same plasticity outcomes with fewer pairings. Second, we chose to use a 'simple' STDP protocol, with a PSP modulated by a single spike, for theoretical clarity and to be consistent with many in vitro corticostriatal findings. A train of spikes may be required in some in vitro situations to induce a reliable effect because conditions are more artificial and less excitable—for example, NMDAR-mediated $Ca^{2+}$ is reduced and AMPAR-mediated $Ca^{2+}$ is strong[35]. In our in vivo situation, it is likely that performing the STDP pairings in the up state enhanced NMDA channel activation, obviating the need for a burst. In the behaving animal situation, the enhanced thalamostriatal depolarization we observed in response to initial reward pairings might progressively increase the likelihood of an up state, thereby leading to greater levels of depolarization during plasticity

induction and a tendency to fire bursts from PSPs. Together, this would mean that plasticity could be induced with fewer pairings, as observed in real-world learning environments.

The eligibility trace set-up by the STDP pairings determines how behaviorally delayed reinforcement signals interact with STDP to induce plasticity. Given that the same reinforcement protocol produced divergent plasticity outcomes depending on the direction of the STDP pairings suggests that qualitatively different types of eligibility traces support potentiation and depression[36]. However, the underlying molecular signals responsible for reinforcement eligibility are unclear. Considering plasticity mechanisms operating in other brain areas it is likely that $Ca^{2+}$ dynamics plays a role in differences between STDP pairing types and resulting eligibility trace mechanisms. Thus, different sources[37], timing in relation to mGluR activation[38] and resulting $Ca^{2+}$ levels[39] following STDP pairings could set two different $Ca^{2+}$ thresholds with different temporal dynamics on which a delayed reinforcement signal could operate. From here, a potential mechanism for potentiation is that elevated $Ca^{2+}$ primes adenylyl cyclase to overcome high phosphodiesterase activity in dendrites. Only with this priming is a later dopamine signal able to activate PKA, and induce potentiation via PKA/DARP-32/PP1 signaling[24, 40]. Detailed analysis of the molecular mechanisms responsible for the eligibility trace is beyond the scope of an in vivo study such as this. Nevertheless, consideration of our study in conjunction with previous in vitro reports could provide testable clues to the molecular mechanisms responsible for this novel form of plasticity.

Previous studies in vitro report that STDP potentiation relies on NMDA-receptor coincidence detector mechanisms[7, 37]. Back-propagating action potentials facilitate $Mg^{2+}$ unblocking of NMDA receptors, which allows postsynaptic $Ca^{2+}$ influx[38]. Consistent with the present results, STDP potentiation was also found to require dopamine signaling[6, 7]. In SPNs of the direct pathway, phasic dopaminergic signaling operating via $D_1$ receptors stimulates adenylyl cyclase activity and the PKA-DARPP32-PP1 pathway[41]. Inhibition of PP1 regulation of NMDA and AMPA receptors critically underlies potentiation[33]. The conditioned light stimulus in the present study is likely to have enhanced and prolonged the phasic dopamine signal originating from midbrain dopamine cell activation, via simultaneous activation of the thalamostriatal innervation of striatal cholinergic interneurons[42]. Synchronous activity in striatal cholinergic neurons promotes striatal dopamine release via activation of nicotinic receptors on dopamine axons[43]. Such synergy would provide a stronger signal for a pro-STDP-potentiation process. Since the activation of cholinergic interneurons invariably precedes a reduction in cholinergic firing, this would result in a phasic dopamine signal arising during this 'pause', thereby removing the 'brake' on plasticity[44] and favoring the induction of LTP[45].

In indirect SPNs, which express $D_2$ receptors, adenosine signaling could induce similar pro-potentiation processes. In the current study, adenosine $A_{2A}$ receptor blockade converted reinforced STDP potentiation into robust depression. This effect is largely consistent with in vitro findings, where antagonism of the $A_{2A}$ receptor disrupts STDP potentiation and promotes LTD in indirect SPNs[7, 46]. Moreover, activation of $D_2$ receptors has consistently been found to be necessary for LTD in indirect pathway SPNs[7, 47]. Conversely, activation of $A_{2A}$ receptors while blocking $D_2$ receptors leads to LTP[7]. These findings are consistent with observations that postsynaptic $A_{2A}$ receptors are located predominantly on indirect pathway SPNs[26], and have an antagonistic interaction with $D_2$ receptors via differential effects on adenylyl cyclase. Postsynaptic $A_{2A}$ receptors signal through adenylyl cyclase to increase cAMP, and promote PKA-DARPP32-PP1 signaling and STDP potentiation processes similar to that

observed with $D_1$ activation[48]. In the present protocol, adenosine signaling is also likely to be coincident with light flash evoked dopamine signaling. Extracellular adenosine levels can be increased by glutamatergic signaling via co-release of ATP and general increases in cellular activity[49]. Hence, it is possible that adenosine increases could occur in response to all aspects of the STDP and reinforcement protocol. Activation of $A_{2A}$ receptors due to the light-induced depolarization would have tipped the balance of intracellular cascades to favor pro-potentiation processes and away from the pro-depression processes associated with $D_2$ activation. In tandem with $D_1$ activation in direct SPNs, these mechanisms could provide the means in vivo for a conditioned reinforcer to induce potentiation in both direct and indirect pathway SPNs following positive timed STDP.

To achieve depression, previous in vitro findings indicate that indirect SPNs require dopaminergic signaling via $D_2$ receptors, and direct SPNs require a relative absence of dopaminergic signaling at $D_1$ receptors[7]. In addition, activation of adenosine $A_{2A}$ signaling would be expected to antagonize the induction of depression in indirect SPNs[7, 46]. And yet, in the present experiment, robust depression was induced with the same reinforcement signals that induced potentiation with positive pairings. Thus negative pairings must induce a distinct eligibility trace that favors the induction of depression in response to reinforcement signaling. The molecular mechanism of such a trace may be associated with coincidence detectors for striatal STDP depression. These involve the PLCβ-mediated release of $Ca^{2+}$ from $IP_3R$-gated stores[37]. This in turn leads to LTD via retrograde endocannabinoid (eCB) signaling[37]. An eligibility trace following negative pairings in vivo might therefore involve prolonged changes in $IP_3$ or PLCβ levels, or $IP_3R$-mediated $Ca^{2+}$ transients. The delayed reinforcement signal from the conditioned light stimulus, involving dopamine, would then interact with this eligibility trace to reduce the efficacy of corticostriatal transmission (depression).

In indirect pathway SPNs, $D_2$ receptor activation has been shown to enhance eCB release[47]. Due to the slow kinetics of mGluR eCB release, the later reinforcement-mediated $D_2$-enhancement could be critical in promoting sufficient eCB release for robust depression at indirect SPNs[50]. In direct pathway SPNs, depression could be mediated by cholinergic interneurons[51]. Activation of $D_2$ receptors on cholinergic interneurons, would induce a pause, relieving the M1-mediated inhibitory cholinergic tone on SPNs, and thereby enhancing L-type $Ca^{2+}$ currents and promoting production of eCBs[52]. A similar mechanism could be initiated by cholinergic pauses driven by excitatory inputs[13] at the time of a fully predicted or, indeed, an omitted reward, when dopamine signaling is itself suppressed[28]. Such mechanisms would provide a means for a conditioned light stimulus to induce depression in both direct and indirect pathway SPNs following negative STDP pairings in vivo.

The correspondence of the electrophysiological findings and the behavioral results suggest a possible functional significance for the components in reinforced corticostriatal plasticity. The input component of the STDP pairings from the motor cortex can be thought of as representing an ongoing action[53]. In this case, positive STDP pairings would arise when the cortical input has contributed to activation of SPNs. Negative STDP pairings would occur when the SPNs fire in response to neurons outside the stimulated cortical ensemble. The later reinforcement components then act to change the representation in the striatum. By supporting potentiation only in the positive pairing cases, causal action representations will be strengthened and novel action-outcome relationships formed. Additionally, the existence of a depression-specific eligibility trace offers further refinement of the action-outcome representation. By depressing active synapses

that could not have contributed to an action yielding reward, this will help increase the signal (positively timed pairings) to noise (negatively timed pairings) ratio. Through such mechanisms, particular representations are biased and may be bound to relevant contextual information and related stimuli. This mechanism would account neatly for Thorndike's Law of Effect where action components associated with subsequent 'satisfaction' in a particular context are more likely to be selected again when the context re-occurs[54].

## Methods

**Animals and electrophysiology.** All procedures were approved by the University of Otago's Animal Ethics Committee. Intracellular recordings were from naive male Long-Evans rats (7–10-week old). Experiments were performed on 205 rats in total. Fifty percent of these experiments yielded neurons. Of these successful experiments, ~1/3 (19% overall) yielded full recordings used in the analysis; 1/2 (24% overall) yielded partial recordings that were interrupted by premature loss of the cell and the remainder (7% overall) yielded recordings that did not meet the criteria of continued cell health throughout the experiment (changes in input resistance or action potential parameters).

Urethane (1.5–2.0 g/kg in 0.9% saline, via intraperitoneal injection (IP); Sigma-Aldrich) was selected as the anesthetic agent due to its established track record for stability in intracellular recording studies investigating in vivo synaptic function[21]. Additional doses were administered as required to maintain slow (0.5–2 Hz) oscillations of the local field potential recorded from the SC.

All surgical procedures, including all electrode implantations, were performed for all protocol groups, regardless of experimental condition. Craniotomies were performed at the following sites, where ipsilateral refers to the left hemisphere used for recording: ipsilateral substantia nigra at AP −4.8 to −5.2, ML +1.7; ipsilateral superior colliculus at AP −6.5, ML +1.5; contralateral medial agranular motor cortex at AP +2, ML −1.6; and a segment (~2 × 3 mm) above the ipsilateral lateral striatum recording site centered at AP +0.5, ML +3. A twisted-pair stimulating electrode (MS303/2-B/SPC, Plastics One) was inserted into the SNc to a depth of 7.4–7.6 mm, and cemented in place. A stainless-steel wire recording electrode was implanted into the SC hole to a depth of 4.5 mm. Finally, a concentric stimulating electrode (Rhodes NEW-100X) was implanted into the contralateral motor cortex site to a depth of 2.2–2.4 mm.

Recordings were made using micropipettes extruded from borosilicate glass capillaries (containing filament; 300127, Harvard Apparatus) of 3 mm in outer diameter, using a vertical puller (PE-22, Narishige). Resistances of the tips of these micropipettes were between 50 and 90 MΩ measured in the brain. Micropipettes were filled with a 1–2% biocytin (Sigma) solution, dissolved in 1 M potassium acetate (Sigma). A silver-chloride wire, made by immersion of bare silver wire into 6% sodium hypochlorite, was placed into the micropipette to interface with the potassium acetate solution. This wire was connected to a headstage (HS-9Ax0.1U, Axon Instruments), which interfaced with a microelectrode amplifier (Axon Axoclamp 900A, Molecular Devices) configured in current-clamp mode. The SC local field potential was recorded via a separate headstage (HS-9Ax1U, Axon Instruments) connected to the same amplifier. For the reference electrode, a silver/silver-chloride pellet (Warner Instruments), was placed subcutaneously approximately 20 mm down the rat's back. The analog recording signal was digitized at 20 kHz (Digidata 1200A, Axon Instruments) and pClamp 10 software (Axon Instruments) was used for protocol configuration, signal display and recording. A locally constructed threshold discriminator was used to trigger events in the SPN down or up state.

Once a cell was impaled and stable, the initiation of an experimental protocol, and the ultimate inclusion of a recording, depended on criteria of cell health: (1) down state membrane potential was more hyperpolarized than −75 mV throughout the experiment; (2) action potential amplitude during current–voltage determinations exceeded 50 mV; (3) membrane dynamics, including inward rectification and a ramp potential during tests of the current–voltage relationship remained intact[21, 22]; (4) input resistance remained stable within a margin of ± 10% throughout the experiment; (5) cortically-evoked PSPs of at least 5 mV in amplitude were present in the five-minute baseline period.

During baseline and test periods, PSPs were always elicited from the onset of the recorded neuron's down state. In its most complete form, the plasticity protocol consisted of a STDP pairing, followed 1 s later by a light flash, and then SNc stimulation a further second later. This full sequence, or variants of it for particular protocol groups, was repeated at approximately 0.1 Hz for 60 events. STDP pairings were comprised of a cortically-induced postsynaptic potential followed or preceded by a single action potential, induced by current injection (0.6–2.2 nA as determined for each cell, for 2.5 ms). Current injection amplitude was increased in 0.1 nA steps to maintain a single action potential if the previous value failed to induce one. The STDP pairings were all performed at the start of the up state, as detected by the threshold discriminator, which caused the STDP sequence frequency to vary between 0.09 and 0.1 Hz. The light stimulus (10 ms duration) was delivered by a white LED (1500 mcd) placed 5 mm in front of the rat's right eye, with the left eye held closed with surgical tape. The contralateral eye was used

because in rats ~90% of visual input to the SC is contralateral[55], and almost all projections to the SC from the SNc are restricted to the same hemisphere[56]. BSR was delivered by a constant-current electrical stimulator (Isolator-10, Axon Instruments; or STG4002-1.6 mA, Multi Channel Systems) and comprised of a biphasic pulse, 500 μs in total width, 500 μA in amplitude, applied at 100 Hz for 500 ms. In experiments involving BSR, once this procedure was performed, no further recordings were made in the animal to rule out potential confounds of prior dopamine release.

SCH 23390 (0.04 mg/kg, in 0.9% NaCl[21]) and KW-6002 (0.8 mg/kg[27]) were administered via an IP catheter ~25 min prior to the STDP pairings. KW-6002 was suspended in a solution of 10% dimethyl sulfoxide (Sigma), 10% polysorbate 80 (Tween-80, Sigma) and 80% saline solution (0.9% NaCl).

**Electrophysiology data analysis.** Analysis was performed with custom functions and scripts written in Matlab R2012b-R2015a (MathWorks). While PSPs were initiated from a largely consistent point at the start of the SPN down state, the responses were still subject to spontaneous membrane potential fluctuations that could contaminate the signal. Hence PSP traces were excluded if they deviated from the group mean by more than 1.5 standard deviations at two standardized points prior to the stimulus artifact and at a typical point after the PSP, when the membrane potential would normally have returned to baseline. On average, 10.3 ± 2.4% of traces were excluded from each experiment, which typically equated to ~30 of 300 traces excluded. Changes in PSPs were measured using the slope of the early component, which most likely represents monosynaptic corticostriatal activation. A linear fit to the PSP was calculated for a 1 ms sliding window over the initial portion of the depolarizing phase, and the maximal slope value was recorded. In cases where the PSP exhibited clear multiple components, the first, and most likely monosynaptic, component was measured. Evidence that the PSP, at least the early component, was monosynaptic includes the fact that latencies were short, with a mean of 5.2 ± 0.6 ms across all groups, and that latencies remained invariant with increasing current intensity.

The striatal response to the light stimulus was analyzed, primarily through the effect of the light event on the latency to up state in the recorded SPN. As the light event was timed to occur 1 s after the STDP event, it could have occurred during any phase of the recorded SPN membrane potential. Only light events that occurred in the down state were used in this analysis—approximately half of them on average. The distribution of these light latencies was compared with latencies to the up state from a random time point created in the preceding down state. Random latencies were only calculated for trials in which a valid light event latency was found. To enable this analysis, thresholds were set automatically for down state, up state, and transition level by the binned distribution of membrane potentials during a two second recorded period around the light event. The down state was defined as the membrane potentials below the twentieth percentile, the up state as above the eightieth percentile, and the transition between states as occurring at the mean of these two potentials. These values were calculated for each light event.

Latencies were modeled as probability distributions for parts of the analysis. Appropriate distributions were selected by using the third-party Matlab function 'allfitdist', which sorts possible distributions by their Bayesian information criterion. A log-logistic distribution was selected for the latencies from the light event to the following up state, based on its goodness of fit to the positive pairings with reinforcement group data, and the Weibull probability distribution for the random latencies to up state, due to its fit with this group's average random data. Cumulative distribution functions were defined using the aforementioned distributions. Kuiper's test was used to determine if the distribution of light latencies was significantly different from the random latencies generated, as it is more sensitive to distribution changes in the tails, where changes would be expected. Comparisons were made between the empirical light latencies and the cumulative distribution functions of the random latencies.

All statistical analyses of electrophysiology data were performed in Prism 6 (GraphPad Software), R[57] (via RStudio), or SPSS Statistics 22 (IBM). The 'lme4' package of R was used to create a linear mixed-effects model (LMM) of PSP slope as a function of the fixed effects protocol and time. As random effects, intercepts for cell identifier were included together with by-cell random slopes for the effect of protocol. The 5-min mean PSP slope bins, of percentage change normalized data, during the 'test' period following the plasticity protocol were modeled. Visual inspection of residual plots did not reveal any obvious deviations from homoscedasticity. Sample size of number of rats used in each group, and the number of trials performed in the behavioral task, were selected on the basis of the observation of robust effects in our many prior related studies. Systematic allocation and not randomization was used to distribute animals to experimental groups. The experimenter was not blinded to experimental condition when applying the protocol but analysis was automated and unbiased.

**Corticostriatal STDP model.** The model consisted of two populations of 11 neurons representing cortical pyramidal cells encoding a particular motor program, and two SPNs, representing a population of SPNs responding primarily to each motor program. Program A cells projected to SPN 1, Program B cells to SPN 2, with a crossover of two cells in each population to the other SPN. The dependent variable measured was the mean synaptic strength of inputs to SPN 1 or SPN 2. To

simulate an action being performed, the neurons of Program A or B were synchronously activated—a synchronous set of spikes was emitted. For initial conditions (Fig. 6d) only one program was used. Actions were emitted continuously with random inter-event time uniformly distributed between 5 and 20 s. Reinforcement was delivered at 1 s from this activation, which consisted of a depolarizing thalamic input and a dopaminergic input. Potentiation was induced when coordinated presynaptic activity caused SPN firing within a ~10 ms time window, and was followed by the conjunction of dopamine and thalamic signals within the time window of the eligibility trace. Depression was induced when the reinforcement signals did not occur, when they arrived sufficiently outside of their time window, or when just dopamine or thalamic signals occurred individually. Both the thalamic input and dopaminergic inputs were modeled using pairs of 'chained' leaky-integrate-and-fire models to replicate the expected 'shape' of the dopaminergic and thalamic responses over time. See Supplementary Methods for mathematical details of the model.

**Behavioral joystick task.** Rats were first implanted with an electrode to support BSR at 7 ± 0.5-week old. Rats were anaesthetized with ketamine (75 mg/kg IP) and domitor (0.5 mg/kg IP), and a stainless-steel twisted-pair stimulating electrode (MS303/2-B/SPC, Plastics One) was implanted into the left SNc (anteroposterior 4.6–5.0 mm, mediolateral 1.8–2.0 mm relative to bregma; dorsoventral 7.5 mm from brain surface). They were housed individually and kept on a reverse day–light cycle, with unrestricted access to food and water, and experiments performed in their night cycle.

Joystick hardware and software methods were as previously described[15], with minor differences in initial joystick training. Rats (n = 17) progressed through three milestones (Supplementary Fig. 3a), and passed the first by achieving 80 hits. Milestones 2 and 3 were passed by achieving performance criteria of a hit-to-miss ratio of >0.3 held for >15 consecutive minutes, and with >80 hits total. A hit was defined as a joystick movement in to the target area and a miss as a movement that does not enter the target area. Rats were automatically progressed to the next milestone within a one-hour experimental session upon meeting the criteria, and were regressed to the prior milestone if no hits were achieved in 5 consecutive minutes. There was a limit of 15 experimental sessions in which to pass milestone 3. Once trained (Supplementary Fig. 3b–d), rats completed two switching experiments with both reward components (baseline). To be included in the analysis (n = 9), they were required to meet a criterion of greater than three blocks completed on average for these two experiments. All criteria were selected based on previous studies[15] as representing reliable learning and at least an animals an average degree of performance in the switching task. The large proportion of animals that did not meet the criteria is indicative of how challenging the joystick task is, especially with a fixed number of sessions in which to learn. Next, a switching session in which the light was removed was completed, and BSR was delivered by itself at the same timing as previously (–light +BSR). The next day behavior was reinstated with the standard reinforcement set in a 15 min pre-session using the milestone 3 target area (Supplementary Fig. 3a). To test for reinstatement during manipulation of the reinforcement components, the hit rate was compared between the first 15 min of the final training session and the reinstatement session (Supplementary Fig. 4). Following a 10 min break, rats then completed a switching session in which BSR was removed, and the light was delivered by itself (+light –BSR).

**Histology and immunohistochemistry.** At the end of either electrophysiological or behavioral experiments, brains were extracted for histology. The rat was perfused intracardially with 4% paraformaldehyde in 0.1 M phosphate buffered solution at pH 7.4. Following perfusion, the brain was extracted and stored in the fixative solution overnight. The brain was then sliced with a vibratome into 80 μM sagittal sections through regions of interest. The positions of implanted electrodes were determined by examining sections through the SNc that were Nissl stained with a 0.03% cresyl-violet solution to enhance the definition of the SNc. The electrode tracks were examined under a dissecting microscope and the position of the most ventral extent of the track was recorded as the electrode tip position (Supplementary Fig. 5).

To recover the recorded cell, sections through the recording region most likely to contain the biocytin-filled neuron were processed. Sections were washed in 1X PBS for 3 ten minute periods, permeabilized in 0.1% triton-PBS solution for 1 h, blocked in a 10% normal donkey serum (9663, Sigma) PBS-triton solution for 20 min, and washed in triton-PBS. Sections were then treated overnight at room temperature with both guinea pig anti-$D_1R$ (1:200, D1R-GP-Af500, Frontier Institute) and rabbit anti-$D_2R$ (1:200, D2R-Rb-Af750, Frontier Institute). The next morning sections were washed in PBS-triton and treated with Cy3-conjugated Donkey Anti-Guinea Pig (1:200, Jackson ImmunoResearch) and Cy5-conjugated Donkey Anti-Rabbit (1:200, Jackson ImmunoResearch) for 2 h. After washing in PBS-triton twice for 20 min, sections were treated with a fluorochrome-conjugated streptavidin (1:100, DyLight 488 Streptavidin, Vector Laboratories) for 2 h, washed in PBS-triton for 20 min, then PBS for a further 20 min, and then mounted on gelatin-dipped slides and allowed to partially dry. Slides were coverslipped with the addition of ProLong Gold anti-fade mount with DAPI incorporated (Life Technologies).

Sections were searched for the DyLight 488 signal of the recorded neuron. Those that could be recovered were characterized as expressing either dopamine $D_1$

or D$_2$ receptors (Fig. 2e; also see Supplementary Methods, and Supplementary Fig. 6)[25]. Anti-D$_1$R and anti-D$_2$R antibodies were validated against Drd1a-EGFP and Drd2-EGFP BAC transgenic mice (Supplementary Fig. 7) and have been previously validated by western blot, analysis in D$_1$R and D$_2$R knockout animals, preabsorption, and exclusive labeling with substance P and enkephalin immunolabeling respectively[25, 58–60]. Positive identification of the recorded neuron was occasionally hindered by intracellular biocytin labeling of multiple SPNs at the recording site, in which case no characterization could be made.

**Code availability**. Matlab code demonstrating the plasticity rule used in the model is freely available at Github: https://github.com/excelsior89/corticostriatal-stdp-rule. The code generates a 2D heat map of the relative change in synaptic strength and runs multiple simulations of reinforcement following a single pre-post pairing, integrating each simulation across time, according to the rule for combining each influence. The code of the full model is part of a larger proprietary system currently under development at the University of Auckland's Bioengineering Institute and is currently not freely distributable. Matlab code for the electrophysiology analysis and behavioral task analysis is available on request from the corresponding author.

**Data availability**. All relevant data supporting the findings of this study are available within the paper (and its Supplementary information file). Raw data can be obtained from the corresponding author on reasonable request.

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

## Acknowledgements

We are very grateful to Professors James Surmeier and Roy Wise for their critical comments on the manuscript. We would like to thank Professor Bernard Balleine for kindly providing Drd2-EGFP and Drd1a-EGFP mice and immunohistochemistry resources. This work was funded by a grant from the Marsden Fund of the Royal Society of NZ (J.N.J.R., W.C.A., and P.R.). S.D.F was supported by the University of Otago PhD scholarship. J.N.J.R. received support from a Rutherford Discovery Fellowship from the Royal Society of New Zealand.

## Author contributions

S.D.F. performed and analyzed the electrophysiological and behavioral experiments and immunohistochemistry, and drafted the manuscript. P.B.R. and M.A.S. designed and characterized the computational model. M.J.B. assisted with behavioral experiments and immunohistochemistry. W.C.A. and P.R. contributed to experimental design and data interpretation. J.N.J.R. designed the experiments, drafted the manuscript, and supervised the study. All authors critically reviewed the manuscript.

## Additional information

**Competing interests:** The authors declare no competing financial interests.

