## [Peer Review File · Nature Communications]

Reviewers' comments:

Reviewer #1 (Remarks to the Author):

In the manuscript entitled "Reinforcement determines the timing dependence of corticostriatal synaptic plasticity in vivo" by Fisher and colleagues, the authors investigated corticostriatal spike-timing-dependent plasticity (STDP) in vivo in the context of behaviorally relevant input signals by intracellular recordings of striatal projection spiny neurons (PSNs) in the rat. They found that bidirectional synaptic plasticity could be induced by pairing pre- and post-synaptic neuronal activity and that it was dependent on dopamine and adenosine signaling. More importantly, they found that in the living animal STDP required sensory reinforcement learning mechanisms that could be obtained by a light flash acting as a conditioned stimulus in a specific time window after the STDP pattern of stimulation. The combination of in vivo electrophysiology with electrical and optical stimulation paradigms to modulate synaptic plasticity and action selection in the living animal is very interesting. The demonstration that an appropriate reinforcement in a specific time window is required to induce corticostriatal potentiation by positive STDP pairings and depression with negative pairings in vivo is also particularly intriguing and very original. Computational analysis and the behavioral test further strengthen the obtained results. I believe that the manuscript reaches a high quality standard and would provide advancement in the knowledge in the field of reinforced corticostriatal plasticity.

As a minor suggestion, the authors might consider to quote the first two papers describing LTD and LTP in the corticostriatal pathway using high frequency stimulation protocols (Calabresi et al, 1992 J.Neurosci; Calabresi et al, 1992 Eur J Neurosci).

Reviewer #2 (Remarks to the Author):

This is a very interesting paper, but it feels preliminary. The authors are putting forward a pretty big concept, but the physiology results are not sufficiently convincing to sustain it.

Comments:

The stimulation frequency for baseline and test pulses should be stated. Representative basal and test PSPs should also be depicted.

If STDP pairings were conducted at 0.1 Hz and only during the UP-state, how did the authors ensure that there was an UP-state every 10 sec? Or was the frequency approximately shifted around 0.1 Hz? This is not clear. Please explain.

The light+BSR positive timing produces a potentiation that is small, but clearly different than those without light. However, no statistical analysis is made relative to basal PSP slope so it has not been established if this is a real potentiation.

The dopamine D1/D2 receptor staining is not convincing. D1 and D2 receptor antibodies are

notoriously non-selective. More low and high magnification images and control experiments demonstrating the specificity of the antibodies (e.g. tested in KO mice) are much needed.

The assumption that the light flash is inducing DA release. However, this has not been directly demonstrated. SN or MFB stimulation at the 1 sec period after pairing should also produce the same effects as the light flash.

Controlled slice electrophysiology experiments demonstrating the sufficiency of dopamine and adenosine in selecting an STDP eligibility trace mimicking in-vivo results would enhance the impact of the study.

Reviewer #3 (Remarks to the Author):

In the manuscript by Fisher et al, they use an in vivo experimental paradigm to investigate the importance of STDP in striatum in the presence and absence of a reinforcing stimulus. Compellingly, the authors use both experiments and modelling to identify important players. This is important work as the in vitro preparations used to investigate STDP in the same system have had difficulties to converge on the relevant sign of STDP as well as the role of timing of the neuromodulatory input. The quality of the science presented is self-evident, and deserves publication. My comments are thus concerned only with improving the clarity of the manuscript and make details more transparent, and thus to make it more widely-accessible and to perhaps draw out additional comments providing insights to readers.

1) It seems that when only an STDP stimulation is given there are almost no plasticity (or only weak LTD), thus the light stimulus is needed and then in turn followed by the BSR. It is argued that the BSR is needed to condition the light stimulus so that it doesn't habituate. In fig 5b,d (the model fig) it seems the authors assume that the light pulse already in the beginning provides a dopamine transient. Are the assumptions thus that the BSR sustains this dopamine elevation caused by SC activation? Thus the role of the BSR is mainly to make sure the dopamine elevation which is elicited by the light pulse stays on? If so, was it tested if replacing the light pulse with the BSR given after 1 sec following the STDP stimulation also gave plasticity. (If BSR came 2 seconds after the STDP that didn't make a difference, but what about 1 second?) Any observations on this? If not tested, please provide any additional thoughts or reasoning regarding this possibility.

2) Intriguingly it seems there might have been some thalamo-striatal plasticity taking place as the probability of an upstate was modified differently in the positive and negative STDP trials. Thus the STDP stimulation in itself, arriving 1 second earlier, seems to be able to modify the thalamo-striatal synapse activated by the light pulse. But perhaps also the dopamine transient resulting from the BST might have been necessary for plasticity in the thalamo-striatal synapse (in addition to prevent habituation in SC). Any thoughts or observations about this?

3) Related to the previous point it was also found that the excitability of the SPNs was

modified depending on the STDP sign (rheobase different in suppl material). If I interpret this correctly it seems the rheobase was not changed significantly following positive STDP, but rheobase was increased following negative STDP pairings. Please discuss this. Potential mechanisms? Parallels to other studies? What could cause excitability to go down? (Any ideas how long this rheobase change was sustained in the experiments?)

4) Seems 60 pairings were needed to get the LTP/LTD results. Isn't this a lot of pairings in an awake, behaving animal? Just read the recent article by Brandalise et al, Nature Communications, 2016, (doi: 10.1038/ncomms13480) and was wondering if something similar is going on in the corticostriatal system? Could the increase in upstate occurrences affected by the light pulse be needed to elicit LTP, and thus could fewer pairings be required in non-anaesthetized animals? Please speculate on this if you feel these exciting findings from the hippocampal system might be relevant also in your system.

5) One small methodological question to be motivated better, or else at least discussed. You use as the postsynaptic stimulus only one AP. Is this a reasonable stimulus from an in vivo point of view? Others (e.g. see Shen et al, 2008) need to use several postsynaptic spikes to elicit plasticity, and also true in many synapses in hippocampus and cortex). Perhaps the use of one AP is compensated for by only pairing the STDP with the occurrence of an upstate in the anaesthetized animal? But isn't it expected that in behaving and wake animals rather several postsynaptic spikes would be involved? Thus perhaps also fewer pairings are needed again? Please motivate and discuss briefly the experimental activation protocol used.

Minor comments:

6) On page 5 on line 17 you cite ref 3 and 12 as examples that positive STDP might give rise to LTD. Please add also a paper from Wickens' group 2011 (doi: 10.1523/JNEUROSCI.3206-11.2011).

7) On e.g. page 13 under the subsection 'Role and nature of the eligibility trace' it is written 6 lines from bottom that 'However, the underlying molecular signals responsible from reinforcement eligibility are unclear'. This is of course true in theory, but still I think one rather could again highlight the ideas emerging from Yagishita et al 2014, where it was suggested that indeed the phosphorylated form of CamKII can function as one important eligibility trace in the presence of PP1 inhibition. Yagishita's ideas were also further elaborated upon, from a quantitative point of view, in a recent modeling study (doi: 10.1371/journal.pcbi.1005080) which can be cited in this section to support this scenario (and further explain why dopamine is most efficient if arriving after the elevation in the NMDA dependent calcium).

8) Finally on page 14, ca 10-12 lines from bottom, it is said that "Since the activation of cholinergic interneurons invariably precedes a reduction in cholinergic firing, this would result in a phasic dopamine signal arising during this 'pause', thereby removing the 'brake' on plasticity (ref 38) and favoring the induction of LTP." Please add another ref (doi: 10.1523/JNEUROSCI.0730-15.2015.) here which has investigated the plausibility this idea

further.

9) Also please deposit the model to a model database (e.g. ModelDB) and provide a link. To write that code, etc 'available on request' is not sufficient after a few year's time. Also if you think your dendritic reconstructions of the SPNs are keeping a high quality (so they can be useful to others), please also deposit those data so the community can use it. (As the purpose in this paper to label SPNs was not really to look at the detailed morphology this point might not be very relevant....fig 1 C seems ok though....)

Response to reviewers

The authors wish to thank all three expert reviewers for their helpful and insightful comments. We have extensively revised our manuscript in line with these comments and believe the manuscript is now considerably improved. We consider each comment below.

Reviewer #1

“As a minor suggestion, the authors might consider to quote the first two papers describing LTD and LTP in the corticostriatal pathway using high frequency stimulation protocols (Calabresi et al, 1992 J.Neurosci; Calabresi et al, 1992 Eur J Neurosci).”

We agree it is valuable to link back to the origins of plasticity findings in these cells. We now reference these papers at the start of the second paragraph of the Introduction.

Reviewer #2

“The stimulation frequency for baseline and test pulses should be stated. Representative basal and test PSPs should also be depicted.”

The stimulation frequencies are stated in Figure 1e, although we have now described these more overtly in the Figure legend.

Representative PSPs, showing the change between baseline and tests periods for each of the experimental conditions, have been added to Figures 1 and 2.

“If STDP pairings were conducted at 0.1 Hz and only during the UP-state, how did the authors ensure that there was an UP-state every 10 sec? Or was the frequency approximately shifted around 0.1 Hz? This is not clear. Please explain.”

The up states were not experimentally manipulated. A threshold discriminator was used to elicit the STDP pairing when the membrane potential reached the next up state after the 10 sec interval. In practice, this meant the interval varied between 10 and 11 seconds, which provided physiological jitter and, more importantly, ensured that the STDP event was only performed at the start of the up state. We have made this point clearer in the Methods section (pg 19 and 20), with the following revised text:

This full sequence, or variants of it for particular protocol groups, was repeated at approximately 0.1 Hz for 60 events... The STDP pairings were all performed at the start of the up state, as detected by the threshold discriminator, which caused the STDP sequence frequency to vary between 0.09 and 0.1 Hz.

“The light+BSR positive timing produces a potentiation that is small, but clearly different than those without light. However, no statistical analysis is made relative to basal PSP slope so it has not been established if this is a real potentiation.”

If we understand the Reviewer correctly, this statistical comparison was described in the legend of Figure 2(c): “time points post protocol vs baseline, estimated effect size = 8.8 ± 3.3 %, $F_{1,46} = 6.69$, $P = 0.013$.” We have illustrated this comparison better by showing it in the Figure itself with an asterisk on a bar across the relevant time points and clarifying the text.

“The dopamine D1/D2 receptor staining is not convincing. D1 and D2 receptor antibodies are notoriously non-selective. More low and high magnification images and control experiments demonstrating the specificity of the antibodies (e.g. tested in KO mice) are much needed.”

This is an important point that was not sufficiently addressed in our original manuscript. We have made a number of changes. First, to provide further examples and support for our prior immunolabeling and cell identification, an additional Supplementary Figure was created. Supplementary Fig. 6 provides images identifying another recorded cell, a close up example of somatic rings, and a secondary-only antibody control.

Second, we performed a new experiment in which we validated the antibodies using *Drd2*-EGFP and *Drd1a*-EGFP BAC transgenic mice. This experiment is described in the Supplementary Notes, and the new Supplementary Fig. 7. The antibodies faithfully identified striatal spiny neurons expressing either D1R or D2R receptors, as defined by the BAC animals.

Third, we have added text to describe prior validation of these antibodies. The antibodies were validated extensively in the original publication describing them, which we referenced in the current manuscript (Narushima et al., 2006). This paper described the production of the antibodies, and validated their specificity by detection of single protein bands by immunoblot; abolition of the immunohistochemical signal by preabsorption; selective perikaryal labelling of D1R in substance P-containing spiny neurons and of D2R in enkephalin-containing spiny neurons; and mutually exclusive labelling patterns in spiny neuron perikarya and neuropils. Additionally, several later publications have independently verified their specificity via Western blot (Hozumi et al., 2008; Moreno et al., 2011) and with striatal D1 and D2 knockout animals (Ikegami et al., 2014). Thus, in the Methods section we have added the following text:

*Anti-D₁R and anti-D₂R antibodies were validated against *Drd1a*-EGFP and *Drd2*-EGFP BAC transgenic mice (Supplementary Fig. 7) and have been previously validated by Western blot, analysis in D₁R and D₂R knockout animals, preabsorption, and exclusive labeling with substance P and enkephalin immunolabeling respectively (Narushima et al., 2006; Ikegami et al., 2014; Hozumi et al., 2008; Moreno et al., 2011).*

Fourth, to the best of our knowledge, one or both of these antibodies have been used for D1/D2 characterization in at least 23 papers (listed in the Appendix below). These papers cover a range of experimental preparations, including rat and mice brain slices, and human tissue culture work. We therefore believe these antibodies have been sufficiently validated in prior publications.

“The assumption that the light flash is inducing DA release. However, this has not been directly demonstrated.”

We have a companion paper now under consideration elsewhere that comprehensively describes the conditioning of light stimuli in the superior colliculus by reinforcement signals. [redacted]

[redacted]

“SN or MFB stimulation at the 1 sec period after pairing should also produce the same effects as the light flash.”

The Reviewer suggests that BSR at 1 second after the positive STDP pairings will lead to potentiation, similar to when the light and BSR are delivered together at 1 and 2 seconds, respectively.

The rationale for our study was to test STDP modulation by a physiological conditioned light stimulus. By pairing the light with BSR we were relying only on BSR's ability to transform the light into a conditioned stimulus that will itself drive physiological levels of phasic dopamine release into the striatum. In addition to a dopamine signal, we have shown that a visual stimulus made salient also provides a simultaneous depolarizing glutamatergic signal to spiny neurons through branched projections from superior colliculus to both the SNc and parafascicular nucleus of thalamus (Schulz et al, *J Neurosci*, 29, 6336-47). Earlier we had shown that depolarization of the spiny neuron is needed in addition to simultaneous activation of the dopamine neurons by BSR to induce LTP (Reynolds et al, *Nature*, 413, 67-70). Thus, a conditioned light stimulus is the signal of choice to induce LTP *in*

vivo because it provides both the necessary simultaneous physiological depolarization and dopamine signal, which is not provided by BSR alone. We therefore do not see value in testing BSR at 1 second because any outcome will not modify our major finding that a conditioned light stimulus can modulate STDP to induce potentiation. We have added a similarly reasoned statement to the discussion on page 13.

“Controlled slice electrophysiology experiments demonstrating the sufficiency of dopamine and adenosine in selecting an STDP eligibility trace mimicking in-vivo results would enhance the impact of the study.”

A number of *in vitro* studies have already demonstrated that signaling via dopamine receptors and adenosine A_{2A} receptors can determine the direction of STDP (e.g. Shen et al., 2008; Yagishita et al., 2014). Our study looked to build on those findings and to demonstrate their efficacy in an *in vivo* environment with physiologically-relevant reinforcement signals.

Reviewer #3

“1) It seems that when only an STDP stimulation is given there are almost no plasticity (or only weak LTD), thus the light stimulus is needed and then in turn followed by the BSR. It is argued that the BSR is needed to condition the light stimulus so that it doesn't habituate. In fig 5b,d (the model fig) it seems the authors assume that the light pulse already in the beginning provides a dopamine transient. Are the assumptions thus that the BSR sustains this dopamine elevation caused by SC activation? Thus the role of the BSR is mainly to make sure the dopamine elevation which is elicited by the light pulse stays on? If so, was it tested if replacing the light pulse with the BSR given after 1 sec following the STDP stimulation also gave plasticity. (If BSR came 2 seconds after the STDP that didn't make a difference, but what about 1 second?) Any observations on this? If not tested, please provide any additional thoughts or reasoning regarding this possibility.”

The Reviewer is correct in stating that the BSR is most likely sustaining the ability of the light stimulus to elicit dopamine, i.e. preventing habituation to the light by developing its status as a conditioned stimulus – this has been added to the discussion on page 13, second paragraph.

The condition in which BSR was delivered alone at 1 sec following positive STDP pairing was not tested in the *in vivo* preparation. We have considered this point in the previous page under Reviewer 2's statement that *“SN or MFB stimulation at the 1 sec period after pairing should also produce the same effects as the light flash.”*

“2) Intriguingly it seems there might have been some thalamo-striatal plasticity taking place as the probability of an upstate was modified differently in the positive and negative STDP trials. Thus the STDP stimulation in itself, arriving 1 second earlier, seems to be able to modify the thalamo-striatal synapse activated by the light pulse. But perhaps also the dopamine transient resulting from the BST might have been necessary for plasticity in the thalamo-striatal synapse (in addition to prevent habituation in SC). Any thoughts or observations about this?”

Yes, as presented in pg 8 of the results the thalamostriatal signal is strengthened only in experiments with STDP pairings that result in potentiation, i.e. positive pairings followed by light + BSR. As this effect was not found with the negative STDP pairing group, followed by the same light + BSR components, the results suggest that there is something important

about the STDP pairings themselves – that their ordering is modulating the thalamostriatal synapses.

The Reviewer asks if the dopamine transient from the BSR could be required for plasticity in the thalamostriatal synapses. The aforementioned finding suggests that dopamine transients by themselves are unlikely to be critical for this thalamostriatal effect, as we demonstrate it still relies on the STDP pairing order. Additionally, we still found an enhanced thalamostriatal effect when D1 receptors were blocked, which suggests that D1 activation is not globally critical. The mechanisms of thalamostriatal plasticity and its interactions with neuromodulator-dependent corticostriatal plasticity would be a very interesting area for future research. We have added a similar statement to the results on pg 9.

“3) Related to the previous point it was also found that the excitability of the SPNs was modified depending on the STDP sign (rheobase different in suppl material). If I interpret this correctly it seems the rheobase was not changed significantly following positive STDP, but rheobase was increased following negative STDP pairings. Please discuss this. Potential mechanisms? Parallels to other studies? What could cause excitability to go down? (Any ideas how long this rheobase change was sustained in the experiments?)”

This is an interesting finding. It might suggest that negative STDP pairings somehow decreased the excitability of the cell, due to different calcium dynamics induced by negative pairing leading to changes in potassium channels that regulate SPN excitability (Day et al., 2008). However, increased rheobase was also found, to a similar extent, in the positive STDP with BSR only (no light) group. This group also experienced clear synaptic depression, but obviously did not undergo the cellular processes to do with negative pairings. All other groups had no significant changes in rheobase. As we now expand in the legend to Supplementary Fig. 1, it appears that cellular processes associated with synaptic depression in general led to changes in excitability. The reasons for this are unclear.

Regarding the change in rheobase over time, we have limited data to answer this question. To avoid confounds we did not want to unnecessarily induce an action potential in the cell during collection of PSPs, thus we only have data points at the start of the experiment, the end of the baseline before STDP pairings, and at the end of the experiment.

We feel that a proper discussion of these points would be outside the scope of our manuscript, as we do not have sufficient data on the excitability changes to make a clear argument. This could be an interesting future area, with a range of separate *in vitro* experiments, ideally using dendritic Ca²⁺ imaging.

“4) Seems 60 pairings were needed to get the LTP/LTD results. Isn't this a lot of pairings in an awake, behaving animal? Just read the recent article by Brandalise et al, Nature Communications, 2016, (doi: 10.1038/ncomms13480) and was wondering if something similar is going on in the corticostriatal system? Could the increase in upstate occurrences affected by the light pulse be needed to elicit LTP, and thus could fewer pairings be required in non-anaesthetized animals? Please speculate on this if you feel these exciting findings from the hippocampal system might be relevant also in your system.”

A primary reasoning behind our decision to go with 60 pairings is that it has been used repeatedly in the *in vitro* literature, so we wanted to create a bridge between the different preparations by keeping this constant. As the reviewer points out, we did find that the light

was more likely to induce an up state during protocols that induced potentiation. However, as this light-induced up state was one second after the STDP pairing it does not seem feasible that it would interact with the STDP pairing in the way reported in the hippocampal *in vitro* system described by Brandalise et al. (2016), i.e. enhancing or inducing dendritic NMDA spikes. Perhaps in awake, behaving animals an enhanced thalamostriatal circuit could increase the likelihood of an up state in general and thereby enhance STDP pairings. In the timing structure we propose, however, we would view the light-induced upstates as having a potential role in enhancing the effect of dopamine on plasticity changes.

Work by Cui et al. (2015) showed that timing-dependent corticostriatal LTP could be biphasic, as determined by the number of pairings (LTP found with 5-15 pairings, no change around 40-50 pairings, and LTP again above 60 pairings). Although this effect may be specific to their *in vitro* preparation, it may reflect unique plasticity processes at corticostriatal synapses. If the latter is true then potentially, we may find the same plasticity outcomes with fewer pairings – we have added a paragraph to discuss this on pg 13/14. Unfortunately, it is difficult to test such hypotheses exhaustively with low yield techniques like *in vivo* intracellular recordings.

“5) One small methodological question to be motivated better, or else at least discussed. You use as the postsynaptic stimulus only one AP. Is this a reasonable stimulus from an in vivo point of view? Others (e.g. see Shen et al, 2008) need to use several postsynaptic spikes to elicit plasticity, and also true in many synapses in hippocampus and cortex). Perhaps the use of one AP is compensated for by only pairing the STDP with the occurrence of an upstate in the anaesthetized animal? But isn't it expected that in behaving and wake animals rather several postsynaptic spikes would be involved? Thus perhaps also fewer pairings are needed again? Please motivate and discuss briefly the experimental activation protocol used.”

Again, we planned to use a ‘simple’ STDP protocol, with a PSP modulated by a single spike, for theoretical clarity and to be consistent with many corticostriatal *in vitro* findings. Once effects were found with this protocol, we retained it for experimental consistency. It appears that a train of spikes is sometimes used in *in vitro* preparations because it is required to achieve an effect. This could be due to *in vitro* environments being more artificial and less excitable – for example, NMDAR-mediated Ca^{2+} is reduced and AMPAR-mediated Ca^{2+} is considerable (Carter and Sabatini, 2004). That said, depolarization is critical in plasticity induction, and hence the Reviewer is likely correct in proposing that performing the STDP pairings in the up state of spiny neurons is a contributing factor to our effect. The up state depolarization would enhance NMDA channel activation. Our results suggest that single PSP and AP pairings can elicit plasticity, in the up state and with reinforcement signals. We would also assume that the system is influenced by strength of inputs and hence agree with the Reviewer that it is likely that plasticity could be found with fewer pairings when PSPs are paired with bursts. We have added this to the paragraph on pg 14.

“6)On page 5 on line 17 you cite ref 3 and 12 as examples that positive STDP might give rise to LTD. Please add also a paper from Wickens' group 2011 (doi: 10.1523/JNEUROSCI.3206-11.2011).”

This reference has been added.

“7)On e.g. page 13 under the subsection ‘Role and nature of the eligibility trace’ it is written 6 lines from bottom that ‘However, the underlying molecular signals responsible from reinforcement eligibility are unclear’. This is of course true in theory, but still I think one rather could again highlight the ideas emerging from Yagishita et al 2014, where it was suggested that indeed the phosphorylated form of CamKII can function as one important eligibility trace in the presence of PP1 inhibition. Yagishita’s ideas were also further elaborated upon, from a quantitative point of view, in a recent modeling study (doi: 10.1371/journal.pcbi.1005080) which can be cited in this section to support this scenario (and further explain why dopamine is most efficient if arriving after the elevation in the NMDA dependent calcium).”

We agree with the reviewer and have added to this discussion on Pg 14/15.

“8) Finally on page 14, ca 10-12 lines from bottom, it is said that “Since the activation of cholinergic interneurons invariably precedes a reduction in cholinergic firing, this would result in a phasic dopamine signal arising during this ‘pause’, thereby removing the ‘brake’ on plasticity (ref 38) and favoring the induction of LTP.” Please add another ref (doi: 10.1523/JNEUROSCI.0730-15.2015.) here which has investigated the plausibility this idea further.”

This reference has been added.

“9)Also please deposit the model to a model database (e.g. ModelDB) and provide a link. To write that code, etc ‘available on request’ is not sufficient after a few year’s time. Also if you think your dendritic reconstructions of the SPNs are keeping a high quality (so they can be useful to others), please also deposit those data so the community can use it. (As the purpose in this paper to label SPNs was not really to look at the detailed morphology this point might not be very relevant....fig 1 C seems ok though....)”

Code demonstrating the plasticity rule has been created in Matlab and placed on Github: <https://github.com/excelsior89/corticostratial-stdp-rule>. The code generates a 2D heat map of the relative change in synaptic strength based on the delay of the light flash and the conditioned dopamine signal induced by the light flash. The script runs multiple simulations of reinforcement following a single pre-post pairing and integrates each simulation across time according to the rule for combining each influence. The code of the full model is part of a larger proprietary system currently under development at the University of Auckland's Bioengineering Institute and is unfortunately not freely distributable. We have updated the manuscript text on pg 24 to reflect the restrictions on availability and the location of the sample Matlab code.

The dendritic morphology was not investigated in detail as the Reviewer suggested, as the identification of the cell body was the desired measure. Hence we do not have suitably high quality reconstructions of these cells.

Appendix

Papers reporting the use of one or both of the D1R and D2R antibodies for D1/D2 characterization.

- Uchigashima, M. et al. Subcellular arrangement of molecules for 2-arachidonoyl-glycerol-mediated retrograde signaling and its physiological contribution to synaptic modulation in the striatum. *J. Neurosci.* 27, 3663–3676 (2007).

- Narushima, M. et al. Tonic enhancement of endocannabinoid-mediated retrograde suppression of inhibition by cholinergic interneuron activity in the striatum. *J. Neurosci.* 27, 496–506 (2007).
- Hozumi, Y. et al. Diacylglycerol kinase beta accumulates on the perisynaptic site of medium spiny neurons in the striatum. *Eur. J. Neurosci.* 28, 2409–2422 (2008).
- Cabello, N. et al. Metabotropic glutamate type 5, dopamine D2 and adenosine A2a receptors form higher-order oligomers in living cells. *J. Neurochem.* 109, 1497–1507 (2009).
- Hozumi, Y., Watanabe, M. & Goto, K. Signaling Cascade of Diacylglycerol Kinase β in the Pituitary Intermediate Lobe: Dopamine D2 Receptor/Phospholipase C β 4/Diacylglycerol Kinase β /Protein Kinase $C\alpha$. *J. Histochem. Cytochem.* 58, 119–129 (2010).
- Moreno, E. et al. Dopamine D1-histamine H3 receptor heteromers provide a selective link to MAPK signaling in GABAergic neurons of the direct striatal pathway. *J. Biol. Chem.* 286, 5846–5854 (2011).
- Fukabori, R. et al. Striatal direct pathway modulates response time in execution of visual discrimination. *Eur. J. Neurosci.* 35, 784–797 (2012).
- Nishizawa, K. et al. Striatal Indirect Pathway Contributes to Selection Accuracy of Learned Motor Actions. *J. Neurosci.* 32, 13421–13432 (2012).
- González-Sepúlveda, M. et al. Cellular distribution of the histamine H3 receptor in the basal ganglia: Functional modulation of dopamine and glutamate neurotransmission. *Basal Ganglia* 3, 109–121 (2013).
- Navarro, G. et al. Cocaine Inhibits Dopamine D2 Receptor Signaling via Sigma-1-D2 Receptor Heteromers. *PLoS One* 8, (2013).
- Dragicevic, E. et al. Cav1.3 channels control D2-autoreceptor responses via NCS-1 in substantia nigra dopamine neurons. *Brain.* 137, 2287–2302 (2014).
- Hernandez, V. S. et al. Dopamine receptor dysregulation in hippocampus of aged rats underlies chronic pulsatile l-Dopa treatment induced cognitive and emotional alterations. *Neuropharmacology* 82, 88–100 (2014).
- Ikegami, M., Uemura, T., Kishioka, A., Sakimura, K. & Mishina, M. Striatal dopamine D1 receptor is essential for contextual fear conditioning. *Sci. Rep.* 4, 3976 (2014).
- Yabuki, Y. et al. Aberrant CaMKII activity in the medial prefrontal cortex is associated with cognitive dysfunction in ADHD model rats. *Brain Res.* 1557, 90–100 (2014).
- Chiken, S. et al. Dopamine D1 receptor-mediated transmission maintains information flow through the cortico-striato-entopeduncular direct pathway to release movements. *Cereb. Cortex* 25, 4885–4897 (2015).
- Eisenhardt, M., Leixner, S., Luján, R., Spanagel, R. & Bilbao, A. Glutamate Receptors within the Mesolimbic Dopamine System Mediate Alcohol Relapse Behavior. *J. Neurosci.* 35, 15523–38 (2015).
- Hozumi, Y. et al. Involvement of diacylglycerol kinase beta in the spine formation at distal dendrites of striatal medium spiny neurons. *Brain Res.* 1594, 36–45 (2015).

- Huck, J. H. et al. De novo expression of dopamine D2 receptors on microglia after stroke. *J. Cereb. Blood Flow Metab.* 35, 1804–11 (2015).
- Takeda, H. et al. Production of monoclonal antibodies against GPCR using cell-free synthesized GPCR antigen and biotinylated liposome-based interaction assay. *Sci. Rep.* 5, 11333 (2015).
- Tyebji, S. et al. Hyperactivation of D1 and A2A receptors contributes to cognitive dysfunction in Huntington's disease. *Neurobiol. Dis.* 74, 41–57 (2015).
- Wood, J. et al. Structure and function of the amygdaloid NPY system: NPY Y2 receptors regulate excitatory and inhibitory synaptic transmission in the centromedial amygdala. *Brain Struct. Funct.* 221, 3373–3391 (2016).
- Naitou, K. et al. Stimulation of dopamine D2-like receptors in the lumbosacral defaecation centre causes propulsive colorectal contractions in rats. *J. Physiol.* 594, 4339–4350 (2016).
- Puighermanal, E. et al. Anatomical and molecular characterization of dopamine D1 receptor-expressing neurons of the mouse CA1 dorsal hippocampus. *Brain Struct. Funct.* 1–15 (2016).
- Rodriguez-Ruiz, M. et al. Heteroreceptor Complexes Formed by Dopamine D1, Histamine H3, and N-Methyl-D-Aspartate Glutamate Receptors as Targets to Prevent Neuronal Death in Alzheimers Disease. *Mol. Neurobiol.* 1–14 (2016).
- Salinas, A. G., Davis, M. I., Lovinger, D. M. & Mateo, Y. Dopamine dynamics and cocaine sensitivity differ between striosome and matrix compartments of the striatum. *Neuropharmacology* 108, 275–283 (2016).

REVIEWERS' COMMENTS:

Reviewer #1 (Remarks to the Author):

The manuscript, that was already an excellent study, has been further improved after the revision to take into account the suggestions and criticisms of the reviews.

Reviewer #2 (Remarks to the Author):

the authors have sufficiently revised the manuscript. I have no further comments.

Reviewer #3 (Remarks to the Author):

All matters suggested in the first review have been revised and discussed satisfactorily, so I feel the revised manuscript on these important findings is now ready for publication.